# Trap-Door Thoracotomy and Clamshell Thoracotomy as Surgical Approaches for Neuroblastoma and Other Thoracic Tumors in Children

**DOI:** 10.3390/cancers16020373

**Published:** 2024-01-15

**Authors:** Benjamin F. B. Mayer, Matthias C. Schunn, Cristian Urla, Jürgen F. Schäfer, Frank Fideler, Felix Neunhoeffer, Martin U. Schuhmann, Steven W. Warmann, Jörg Fuchs

**Affiliations:** 1Department of Pediatric Surgery and Pediatric Urology, University Children’s Hospital Tübingen, Hoppe-Seyler Straße 3, 72076 Tübingen, Germany; matthias.schunn@med.uni-tuebingen.de (M.C.S.); cristian.urla@med.uni-tuebingen.de (C.U.); steven.warmann@med.uni-tuebingen.de (S.W.W.); joerg.fuchs@med.uni-tuebingen.de (J.F.); 2Division of Pediatric Radiology, Department of Diagnostic Radiology, University Hospital Tübingen, Hoppe-Seyler Straße 1, 72076 Tübingen, Germany; juergen.schaefer@med.uni-tuebingen.de; 3Department of Anesthesiology and Intensive Care Medicine, University Hospital Tübingen, Hoppe-Seyler Straße 1, 72076 Tübingen, Germany; frank.fideler@med.uni-tuebingen.de; 4Department of Pediatric Cardiology, Pulmonology and Intensive Care Medicine, University Children’s Hospital Tübingen, Hoppe-Seyler Straße 3, 72076 Tübingen, Germany; felix.neunhoeffer@med.uni-tuebingen.de; 5Division of Pediatric Neurosurgery, Department of Neurosurgery, University Hospital Tübingen, Hoppe-Seyler Straße 1, 72076 Tübingen, Germany; martin.schuhmann@med.uni-tuebingen.de

**Keywords:** neuroblastoma, solid thoracic tumors, trap door, clamshell, children, neuromonitoring, angiography

## Abstract

**Simple Summary:**

Solid thoracic tumors in children are rare, with neuroblastoma being the most commonly diagnosed tumor entity. A safe and complete tumor resection is often not possible with a unilateral cervical approach, sternotomy, thoracotomy, or bilateral thoracotomy (single or staged) in certain localizations of the thorax, such as the cervicothoracic junction, the posterior mediastinum, or bilateral dorsal thoracic tumors. In adults, trap-door thoracotomy has been established and evaluated as a safe surgical approach for tumors of the cervicothoracic junction and clamshell thoracotomy as a safe surgical approach for posterior mediastinal or bilateral dorsal thoracic tumors. The aim of this study was to evaluate the surgical and oncologic implications of trap-door thoracotomy and clamshell thoracotomy in children with solid tumors in these specific localizations. These findings may have a positive impact on the quality of care and patient safety for children with solid thoracic tumors.

**Abstract:**

Solid tumors of the cervicothoracic junction, the posterior mediastinum, or bilateral dorsal thoracic tumors represent a challenge in pediatric surgical oncology. The aim of this study was to evaluate trap-door thoracotomy and clamshell thoracotomy as surgical approaches. A single-center retrospective study of children with solid tumors in these specific localizations was performed. From 2015 to 2023, 26 children (17 girls; 9 boys) were treated at a median age of 54 months (range 8–229). Tumor resection was performed for neuroblastoma (*n* = 11); metastatic disease (*n* = 7); malignant rhabdoid tumor (*n* = 4); Ewing sarcoma (*n* = 1); inflammatory myofibroblastic tumor (*n* = 1); rhabdomyosarcoma (*n* = 1); and neurofibroma (*n* = 1). The surgical goal of macroscopic complete excision was achieved in all of the 14 children who underwent trap-door thoracotomy and in 11 of the 12 children who underwent clamshell thoracotomy. There were no major complications. At a median follow-up of 8 months (range 0–60), the disease was under local control or in complete remission in 66.7% of the children. In conclusion, surgical resection of solid tumors of the cervicothoracic junction in children can be performed safely and successfully with trap-door thoracotomy and with clamshell thoracotomy for posterior mediastinal or bilateral dorsal thoracic tumors.

## 1. Introduction

Solid thoracic tumors in children are rare, with neuroblastoma being the most common malignancy [1]. Surgically, the resection of tumors located at the cervicothoracic junction, posterior mediastinum, or bilateral dorsal thoracic tumors is challenging because they often cannot be fully visualized through a unilateral cervical approach, sternotomy, thoracotomy, or bilateral thoracotomy (single or staged) [1,2]. However, the adequate visualization of these tumors is essential to avoid injury to major nerves and vessels. For example, in the case of retrocardiac tumors or tumors infiltrating the pericardium or diaphragm, tumor resection through sternotomy or thoracotomy requires extensive manipulation of the heart with the risk of rapid iatrogenic cardiac arrest due to decreased cardiac preload and subsequent reduced cardiac output. In neuroblastoma, the surgical goal is to achieve complete macroscopic excision of more than 95% of the visible tumor [3]. For the resection of thoracic sarcomas, malignant rhabdoid tumors, or pulmonary metastases, complete microscopic tumor resection is mandatory [4,5,6,7]. This requires the en bloc resection and often reconstruction or replacement of affected nerves and vessels. Because of the associated morbidity and the need for successful resection to achieve a positive oncologic outcome, alternative surgical approaches for the resection of tumors located at the cervicothoracic junction, in the posterior mediastinum, or bilateral dorsal thoracic tumors should be considered in the surgical strategy.

For the resection of solid tumors of the cervicothoracic junction in adult patients, a combination of supraclavicular incision, sternotomy, and anterior thoracotomy (trap-door thoracotomy) has been shown to allow for complete resection with low morbidity [8,9,10]. In adult cardiothoracic surgery, bilateral anterior thoracotomy (clamshell thoracotomy) has been established as a surgical approach to the posterior mediastinum and to bilateral dorsal thoracic tumors. It allows for the complete resection of large mediastinal tumors or metastases in both lungs with low morbidity [11,12]. To date, trap-door thoracotomy and clamshell thoracotomy have only been evaluated in a small number of children as surgical approaches for the resection of tumors in these specific thoracic localizations [13,14,15]. In particular, the indications for both procedures and the requirements and expansion of preoperative imaging have not been extensively evaluated. Therefore, the aim of this study was to provide a comprehensive evaluation of the indications for both procedures, preoperative imaging requirements, surgical outcomes, as well as oncologic implications.

## 2. Materials and Methods

### 2.1. Study Design

A retrospective single-center cohort study of consecutive children who underwent the resection of solid cervicothoracic tumors at our institution between 2015 and 2023 was performed. Patients were included if they had undergone surgical resection of solid thoracic tumors by trap-door or clamshell thoracotomy. The standard preoperative diagnostic workup consisted of cross-sectional imaging by magnetic resonance imaging (MRI) or computed tomography (CT) in all patients. It was supplemented by ultrasound for peripheral nerve involvement, cerebral angiography for cervical vascular compression, and spinal angiography for posterior mediastinal tumors to identify the artery of Adamkiewicz to avoid spinal cord ischemia [16]. Vascular encasement was defined as tumor contact with more than 50% of the vessel circumference [17].

Trap-door thoracotomy was indicated for cervicothoracic tumors and clamshell thoracotomy for posterior mediastinal or bilateral dorsal thoracic tumors for which safe and complete resection was not feasible by a unilateral cervical approach, sternotomy, or thoracotomy after the evaluation of preoperative imaging. In all patients, a thoracic epidural catheter was placed immediately prior to surgery under general anesthesia for postoperative pain management. Data were retrieved from hospital records and stored in a computerized database (Microsoft Excel, Version 1808, Microsoft Corporation, Redmond, WA, USA).

### 2.2. Trap-Door Thoracotomy

A trap-door thoracotomy is performed in a modified fashion as previously described (Figure 1 and Appendix A) [8,9,10]. General anesthesia is established with a double-lumen or single-lumen endotracheal tube if the lung apex is involved. The patient is placed in the supine position with a shoulder roll, the arm ipsilateral to the lesion outstretched, and the head turned to the contralateral side. A supraclavicular transverse incision is made toward the midportion of the sternum, leaving the clavicle intact, and then extended inferiorly in the midline of the sternum to the intercostal space where the inferior end of the tumor extends. The incision is then completed laterally to the anterior axillary line. The intercostal muscles and, if necessary, the pectoralis muscles are divided and the internal mammary vessels are transected after ligation. The retrosternal space is dissected bluntly before dividing the sternum with a sternal saw. The osteomuscular flap, the so-called “trap door”, can then be lifted laterally to expose the surgical field. Dissection begins with the mobilization of the thymus and the identification of the major vessels and nerves of the upper mediastinum. After the completion of tumor resection, a chest tube is placed and the sternum is reapproximated with steel wires in older patients and non-absorbable sutures in infants. Pericostal sutures are placed to reapproximate the ribs. The pectoralis and sternocleidomastoid muscles and the subcutaneous and dermal layers of the incision are reconstructed with absorbable sutures.

### 2.3. Clamshell Thoracotomy

A clamshell thoracotomy is performed as a modification of the technique first described by Bains et al. (Figure 2 and Appendix A) [11]. General anesthesia is established with a double-lumen endotracheal tube. The patient is placed in the supine position with both arms abducted and a roll placed under the mid-chest. A bilateral incision is made below the nipples, sparing the mammary glands, following the course of the intercostal spaces with a connecting transverse incision over the lower part of the sternum. The intercostal muscles are divided and the internal mammary vessels are transected after ligation. The retrosternal space is dissected bluntly before dividing the sternum with a sternal saw. The cranial osteomuscular flap is lifted with retractors to expose the surgical field. After the completion of tumor resection, bilateral chest tubes are placed and connected with a Y-adapter. The sternum is reapproximated with steel wires in older patients and non-absorbable sutures in infants, and the ribs with pericostal sutures. The subcutaneous and dermal layers of the incision are reconstructed with absorbable sutures.

### 2.4. Demographic and Clinical Characteristics

Demographic characteristics included age and sex. Preoperative clinical characteristics included tumor entity, age at diagnosis, current tumor site and primary tumor manifestation, disease phase (primary/secondary/tertiary), disease stage, tumor genetics, previous surgery, neoadjuvant chemotherapy and radiotherapy, type of preoperative imaging (MRI/CT/ultrasound of peripheral nerves/cervical or spinal angiography), affected structures, preoperative symptoms, age at surgery, and surgical approach (trap-door/clamshell).

### 2.5. Surgical and Oncological Data

The primary outcomes of interest were operative time, resection with neurosurgery and use of intermittent neuromonitoring, duration of mechanical ventilation, duration of chest tubes, length of ICU and hospital stay, postoperative complications, and resection status. For neuroblastoma resections, complete macroscopic resection was defined as resection of >95% of the visible tumor [18]. For all other tumor entities, complete microscopic resection was reported if the resection margins were tumor-free based on histopathologic evaluation. Postoperative complications were classified according to the Clavien–Dindo classification [19]. Oncologic data included the administration of adjuvant chemotherapy or radiotherapy and oncologic outcome at last follow-up.

### 2.6. Statistical Analysis

Patients who underwent a resection of cervicothoracic junction tumors, posterior mediastinal tumors, or bilateral dorsal thoracic tumors by trap-door thoracotomy or clamshell thoracotomy were analyzed using descriptive statistics (Microsoft Excel, Version 1808, Microsoft Corporation, Redmond, WA, USA).

## 3. Results

Between 2015 and 2023, a total of 73 children underwent a resection of thoracic tumors in our department. In 26 children (17 girls, 9 boys), the tumors were located at the cervicothoracic junction, in the posterior mediastinum, or were bilateral dorsal thoracic tumors (Table 1). The majority of patients underwent tumor resection by trap-door thoracotomy or clamshell thoracotomy for neuroblastoma (46.2%) or malignant rhabdoid tumor (19.2%). N-MYC amplification was found in two of the twelve neuroblastomas and SMARCB1 amplification in two of the five malignant rhabdoid tumors. Resection of primary thoracic tumors was performed in 19 children and resection of thoracic metastases was performed in 7 children. Metastasectomy was performed as primary treatment in three patients and as secondary or tertiary treatment after relapse in four patients. Major structures were involved in all but two patients with metastatic disease confined to the lung parenchyma. Nearly half of the patients had preoperative symptoms, most of which were attributed to structures affected by the tumor, such as Horner’s syndrome, limb paresis, pain, or lymphedema in tumors of the cervicothoracic junction. Except for one patient with neurofibroma and one patient with ganglioneuroma, all patients received neoadjuvant chemotherapy. Preoperative imaging was supplemented by specialized imaging in 11 patients. After tumor resection, chest tubes were placed and all patients were transferred to the pediatric intensive care unit (ICU) after surgery. In two patients in the trap-door thoracotomy group and one patient in the clamshell thoracotomy group, ICU stay was longer than 7 days due to the need for intensive ventilation therapy for diaphragmatic paresis and/or pneumonia or dystelectasis. Histopathologic examination revealed ganglioneuroma in 5 of 12 patients with neuroblastoma. Therefore, only a total of 18 patients received adjuvant chemotherapy (Table 2). Neoadjuvant immunotherapy was administered to three patients. A total of three patients were lost to follow-up due to non-response to follow-up requests. At a median follow-up of 8 months (0–60), the disease was in complete remission or under local control in 66.7% of patients.

### 3.1. Trap-Door Thoracotomy

In the trap-door group, two patients had previously undergone tumor resection by thoracoscopy and thoracotomy. The thoracotomy scar could be integrated into the trap-door incision. Trap-door thoracotomy was performed for tumors located at the cervicothoracic junction (*n* = 10) and in the superior mediastinum (*n* = 4). In the majority of patients (7 of 14), tumor resections were performed by trap-door thoracotomy in conjunction with neurosurgery (Table 2). Complete microscopic resection or, in the case of neuroblastoma and neurofibroma, complete macroscopic excision was achieved in all patients. The majority of postoperative complications were minor and mostly due to resection near nerve structures; they included Horner’s syndrome or diaphragmatic paresis. As a major complication, a large chest wall hematoma was surgically evacuated in one patient after trap-door thoracotomy. Of the four patients with tumor recurrence, three patients had local tumor recurrence.

### 3.2. Clamshell Thoracotomy

Half of the patients (6 of 12) underwent clamshell thoracotomy for the resection of metastatic disease. In three patients, a previous resection of a thoracic tumor had been performed by thoracotomy. The thoracotomy scar was integrated into the clamshell incision in all patients. All tumors located in the posterior mediastinum (*n* = 8) or bilateral dorsal thorax (*n* = 4) were resected by clamshell thoracotomy. In three patients, bradycardia and hypotension occurred due to pericardial retraction for tumor resection. Intraoperative bronchial injury occurred in one patient requiring reconstruction and intraoperative bronchoscopy. Complete macroscopic excision was achieved in all but one patient. In this patient, the progression of lung metastases from osteosarcoma was found intraoperatively, and, therefore, only biopsies were taken for further molecular diagnosis (Table 2). Pneumothorax and pleural effusion occurred as major complications in three patients after clamshell thoracotomy and required secondary chest tube placement under general anesthesia. At the last follow-up, three patients who underwent clamshell thoracotomy for metastasectomy had died.

**Table 2 cancers-16-00373-t002:** Study outcomes.

Variable	All (*n* = 26)	Trap-Door (*n* = 14)	Clamshell (*n* = 12)
Operative time [minutes] *	197 (104–424)	207 (122–424)	179 (104–381)
Resection with neurosurgery [*n*]	7 (26.9%)	7 (26.9%)	0
Intermittent neuromonitoring [*n*]	8 (30.8%)	8 (30.8%)	0
Resection status			
MCE (neuroblastoma) **	11 (42.3%)	7 (26.9%)	4 (15.4%)
R0 resection (other tumors)	14 (53.9%)	7 (26.9%)	7 (26.9%)
MCE (all tumor entities) **	25 (96.2%)	14 (53.8%)	11 (42.3%)
Biopsy	1 (3.9%)	0	1 (3.9%)
Duration of ventilation [days] *	1 (0–6)	1 (0–5)	1 (0–6)
Duration of chest tube [days] *	6 (1–14)	6 (3–11)	6 (1–14)
Length of ICU stay [days] *	2 (1–12)	3 (1–12)	2 (1–8)
Length of hospital stay [days] *	10 (6–35)	9 (6–22)	14 (7–35)
Complications [*n*] ***	17 (54.2%)	9 (25%)	8 (29.2%)
I. Horner syndrome	7 (26.9%)	7 (26.9%)	0
Diaphragm paresis	6 (23.1%)	5 (19.2%)	1 (3.9%)
Pleural effusion	3 (11.5%)	2 (7.7%)	1 (3.9%)
Paresis after nerve resection	3 (11.5%)	3 (11.5%)	0
II. Pneumonia or C. diff. infection	2 (7.7%)	2 (7.7%)	0
III. b. Pneumothorax	2 (7.7%)	0	2 (7.7%)
Pleural effusion	1 (3.9%)	0	1 (3.9%)
Chest wall hematoma	1 (3.9%)	1 (3.9%)	0
Pathology result differs from biopsy [*n*]	8 (30.8%)	6 (23.1%)	2 (7.7%)
Adjuvant chemotherapy [*n*]	18 (69.2%)	8 (30.8%)	10 (38.5%)
Adjuvant radiotherapy [*n*]	5 (19.2%)	3 (11.5%)	2 (7.7%)
Follow-up [months] *	8 (0–60)	10 (5–21)	7 (0–60)
Lost to follow-up	9 (34.6%)	4 (15.4%)	4 (15.5%)
Tumor recurrence [*n*]	7 (38.8%)	4 (22.2%)	3 (16.7%)
Oncological outcome [*n*]			
Complete remission	6 (33.3%)	4 (22.2%)	3 (16.7%)
Local control	5 (27.8%)	4 (22.2%)	1 (5.6%)
Progress	2 (11.1%)	1 (5.6%)	1 (5.6%)
Death	4 (22.2%)	1 (5.6%)	3 (16.7%)

ICU: Intensive care unit, MCE: complete macroscopic excision, R0: complete microscopic resection, C. diff.: Clostridium difficile. * Data are expressed as median (range). ** Complete macroscopic excision of >95% of visible tumor [3]. *** Classification according to Clavien–Dindo [19].

## 4. Discussion

In many conditions of solid thoracic tumors in children, surgery plays a key role in local tumor control. In various entities, the completeness of resection impacts oncological outcomes [5,7,20,21,22]. Tumors at the cervicothoracic junction, in the posterior mediastinum, or bilateral thoracic tumors are in close proximity to vital structures and therefore resection is associated with high perioperative morbidity [15]. Because successful resection often requires the reconstruction or replacement of these structures, alternative surgical approaches must be considered when planning the surgical strategy. This study evaluated trap-door thoracotomy and clamshell thoracotomy for the surgical resection of neuroblastoma and other solid cervicothoracic, posterior mediastinal, or bilateral dorsal thoracic tumors in children. The surgical management of tumors in these specific thoracic localizations in children requires specialized preoperative imaging and an established interdisciplinary pediatric surgical–neurosurgical collaboration to achieve complete tumor resection while preserving nerve function and organ perfusion [23]. Trap-door thoracotomy and clamshell thoracotomy are also feasible as secondary procedures after previous thoracotomy, integrating existing scars into the incision line and thus avoiding additional thoracic trauma. In combination with adequate pain management by thoracic epidural analgesia, both approaches allow for early recovery after surgery [24]. In this study, we demonstrated that trap-door thoracotomy with intermittent neuromonitoring allows for the safe and successful resection of solid tumors of the cervicothoracic junction. For large mediastinal and/or bilateral lung tumors, we have shown that resection can be performed successfully with low morbidity using clamshell thoracotomy.

The patient population studied in this study is comparable in terms of epidemiologic and clinical characteristics to those previously studied. As described in other studies, neuroblastoma and other neurogenic tumors were the most common tumor entity [13,14,15,25]. Our cohort included a relatively large number of five malignant rhabdoid tumor patients compared to one previously reported in the literature [25]. The indication for either surgical approach was based solely on the localization of the tumor, as the first descriptors suggest [8,11]: trap-door thoracotomy was performed for tumors located at the cervicothoracic junction or superior mediastinum, and clamshell thoracotomy was performed for posterior mediastinal and bilateral dorsal thoracic tumors. The challenge of the surgical management of solid tumors in anatomically complex localizations of the thorax was underscored in this study by the high number of major structures involved by the tumor and the associated symptomatology. Given this challenging anatomic situation, 11 patients required additional specialized imaging to plan the surgical strategy accordingly. In the trap-door group, peripheral nerve ultrasound was performed the day before surgery to better understand the course of the nerve and its relationship to the tumor in three patients [26]. In one patient with tumor thrombosis of the internal jugular vein and upper extremity lymphedema, cerebral angiography was performed to ensure adequate venous drainage of the head. In the clamshell group, preoperative spinal angiography identified the artery of Adamkiewicz in seven patients. It could be spared intraoperatively, preventing spinal cord ischemia [16]. Although this preoperative imaging modality has not been described in previous studies of posterior mediastinal tumor resection via trap-door or clamshell thoracotomy, we strongly recommend performing spinal angiography preoperatively and adjusting the surgical strategy accordingly.

In this study, tumors of the cervicothoracic junction could be safely resected in children with low morbidity by trap-door thoracotomy. All tumors involving major nerve structures were resected together with a neurosurgeon using intermittent neuromonitoring, as shown in Appendix A and also suggested by Sala et al. [27]. As shown in Appendix A, the subclavian vein was resected and replaced with a graft due to tumor infiltration. This would not have been possible with a unilateral cervical approach or thoracotomy. The surgical goal of complete resection, or complete macroscopic excision in the case of neurogenic tumors, was achieved in all cases, which we consider particularly important since nine cases were high-risk neuroblastomas [22]. Consistent with other studies investigating trap-door thoracotomy in children, perioperative morbidity was low, with the surgical evacuation of a chest wall hematoma being the only complication requiring an intervention [13,14,15]. Most of the minor complications were due to tumor resection at or near nerve structures. Although not structurally documented, no shoulder girdle instability was observed due to the preservation of sternoclavicular articulation and pectoralis innervation with trap-door thoracotomy [28]. In one patient, shown in Figure 1, trap-door thoracotomy was performed twice for near-total tumor resection and secondary complete resection after neoadjuvant chemotherapy without musculoskeletal sequelae. Although a transmanubrial approach, as described by Sauvat et al., has also been suggested for the resection of thoracic inlet tumors, we found this approach not feasible in our cohort as most tumors extended beyond the subclavian vein [29]. Consistent with other studies and despite the high number of patients with prognostically poor malignant rhabdoid tumors in our patient cohort, the majority of patients were alive at last follow-up [13,14,15]. Tumor recurrence was treated surgically in two patients and with proton therapy in another patient. Another patient with tumor progression is currently being scheduled for surgical resection. In light of the results of this study and the data available in the literature, we recommend trap-door thoracotomy with neurosurgery and neuromonitoring in case of major nerve involvement as the primary surgical approach for solid tumors of the cervicothoracic junction in children.

For posterior mediastinal and bilateral dorsal thoracic tumors, we demonstrated that resection can be performed successfully with low morbidity using the clamshell thoracotomy approach. With a median operative time of approximately three hours and no major complications, all tumors could be resected quickly and safely, which is consistent with other studies [12,22]. For bilateral dorsal thoracic tumors, we found that the lower lobes could be well dissected because the pulmonary ligament was well accessible through the clamshell thoracotomy. For retrocardiac tumors infiltrating the pericardium or diaphragm, successful resection was possible even in hemodynamically unstable patients because the heart could be retracted cranially after complete dissection from the diaphragm without critically reducing cardiac preload and output. We consider this to be the major advantage of clamshell thoracotomy in comparison to sternotomy and unilateral or bilateral thoracotomy. Another advantage of clamshell thoracotomy is that it can be extended distally to allow access to the thoracic and abdominal cavities, thus facilitating resection of thoracoabdominal tumors in a single-stage approach (Figure 1) [23]. Some authors have proposed a staged bilateral thoracotomy for the resection of pulmonary metastases [1]. This is associated with a longer hospital stay and a longer time to administration of chemotherapy than a single-stage approach [2]. As we have demonstrated with clamshell thoracotomy, metastasectomy can be performed with low morbidity and moderate length of hospital stay in a single-stage approach. This may reduce the time to adjuvant chemotherapy, which is particularly important in metastatic disease [30].

The results of this study should be interpreted within its limitations. These limitations include the small number of patients and the heterogeneity of the study population due to the single-center, retrospective design. However, solid tumors of the cervicothoracic junction, the posterior mediastinum, or bilateral dorsal thoracic tumors are rare in children, and only in a small number of patients is complete resection and sparing of adjacent nerves and vessels not possible with conventional surgical approaches. Therefore, a prospective study evaluating trap-door thoracotomy and clamshell thoracotomy for the resection of tumors in these specific thoracic localizations would need to be performed as a multicenter study in collaboration with other centers specialized in pediatric surgical oncology.

## 5. Conclusions

Neuroblastoma and other solid thoracic tumors can be successfully and safely resected using trap-door thoracotomy for cervicothoracic tumors and clamshell thoracotomy for posterior mediastinal or bilateral dorsal thoracic tumors. When resecting thoracic neuroblastoma, trap-door thoracotomy and clamshell thoracotomy allow for the nerve- and vessel-sparing and safe resection of retrocardiac tumor segments without compromising cardiac outflow. For the resection of sarcoma or other solid thoracic tumor entities, trap-door thoracotomy and clamshell thoracotomy allow for en bloc tumor resection and for the reconstruction or replacement of vital structures. In anatomically complex cases of tumor infiltration, preoperative imaging should be supplemented by specialized imaging to identify structures at risk and minimize surgical complications. Resections should be performed in specialized pediatric surgical oncology centers to achieve high rates of complete tumor resection with good surgical, functional, and oncologic outcomes.

## Figures and Tables

**Figure 1 cancers-16-00373-f001:**
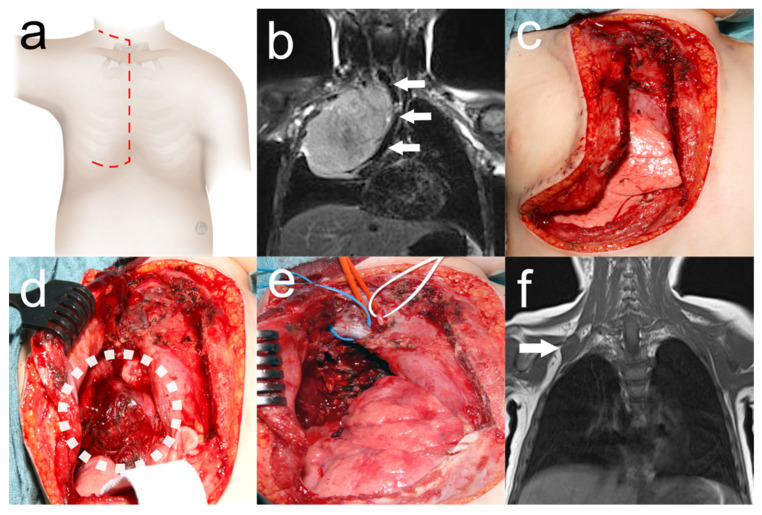
Trap-door thoracotomy for resection of Ewing sarcoma of the cervicothoracic junction. Two-year-old girl who underwent tumor resection by trap-door thoracotomy ((**a**) red dashed line: surgical incision) with respiratory distress due to Ewing’s sarcoma. Preoperative MRI (**b**) showing a tumor of the right cervicothoracic junction causing mediastinal shift with compression of the trachea (white arrows). Urgent tumor resection was performed, leaving the mammary gland intact (**c**). Tumor exposure (white dotted circle) after lifting the osteomuscular flap (**d**). Surgical site (**e**) after near-total tumor resection with the right brachiocephalic vein (blue loop), the right brachiocephalic trunk (red loop), and the phrenic nerve (white loop). Tumor remnant adjacent to the second loop (white arrow) on MRI (**f**) after adjuvant chemotherapy. Secondary complete tumor resection was again performed by trap-door thoracotomy.

**Figure 2 cancers-16-00373-f002:**
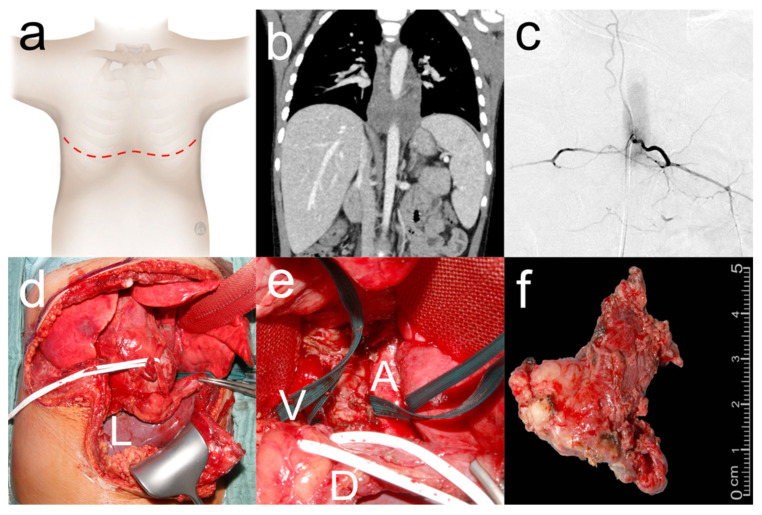
Extended clamshell thoracotomy for resection of thoraco-abdominal neuroblastoma. Five-year-old boy who underwent clamshell thoracotomy ((**a**) red dashed line: surgical incision) for resection of a thoraco-abdominal neuroblastoma. Preoperative CT (**b**) showed complete encasement of the aorta. Preoperative spinal angiography (**c**) showed artery of Adamkiewicz arising from the left aorta at the 12th thoracic vertebral body. Clamshell thoracotomy (**d**) was extended distally in the midline (L: liver) and the pericardium and diaphragm were incised to expose a retrocardiac tumor as retraction of the heart resulted in bradycardia and hypotension with the need for intermittent CPR. Surgical site (**e**) with tumor between aorta (A) and superior vena cava (V) with phrenic nerve (white loop) and diaphragm (D) retracted anteriorly. Tumor (**f**) after gross total resection.

**Table 1 cancers-16-00373-t001:** Demographic and clinical characteristics.

Variable	All (*n* = 26)	Trap-Door (*n* = 14)	Clamshell (*n* = 12)
Sex [*n*]			
Female	17 (65.4%)	8 (30.8%)	9 (34.6%)
Male	9 (34.6%)	6 (23%)	3 (11.5%)
Median age at diagnosis [months] (range)	51 (4–205)	50 (4–151)	42 (11–205)
Tumor entity [*n*]			
Neuroblastoma	12 (46.2%)	6 (23.1%)	6 (23.1%)
Malignant rhabdoid tumor	4 (15.4%)	4 (15.4%)	0
Ewing sarcoma	1 (3.9%)	1 (3.9%)	0
Rhabdomyosarcoma	1 (3.9%)	0	1 (3.9%)
Inflammatory myofibroblastic tumor	1 (3.9%)	1 (3.9%)	0
Metastatic disease	7 (26.9%)	1 (3.9%)	6 (23.1%)
Osteosarcoma	3 (11.5%)	1 (3.9%)	2 (7.7%)
Neuroblastoma	1 (3.9%)	0	1 (3.9%)
Malignant rhabdoid tumor	1 (3.9%)	0	1 (3.9%)
Hepatoblastoma	1 (3.9%)	0	1 (3.9%)
Renal clear cell sarcoma	1 (3.9%)	0	1 (3.9%)
Tumor localization [*n*]			
Mediastinum	12 (46.2%)	4 (15.4%)	8 (29.2%)
Cervicothoracic junction	6 (23.1%)	6 (23.1%)	0
Thoracic inlet	4 (15.4%)	4 (15.4%)	0
Bilateral dorsal thorax	4 (15.4%)	0	4 (15.4%)
Affected structures [*n*]			
Major vessels encased	21 (80.8%)	12 (46.2%)	9 (34.6%)
Nerves and neuroforamina	16 (61.5%)	12 (46.2%)	4 (15.4%)
Trachea and bronchi	7 (26.9%)	3 (11.5%)	4 (15.4%)
Esophagus	7 (26.9%)	2 (7.7%)	5 (19.2%)
Pericardium	5 (19.2%)	1 (3.9%)	4 (15.4%)
Preoperative symptoms [*n*]	11 (42.3%)	8 (30.8%)	3 (11.5%)
Horner syndrome	3 (11.5%)	3 (11.5%)	0
Respiratory insufficiency	3 (11.5%)	1 (3.9%)	2 (8.3%)
Paresis upper limb	2 (7.7%)	2 (7.7%)	0
Pain upper limb	2 (7.7%)	2 (7.7%)	0
Upper inflow congestion	2 (7.7%)	1 (3.9%)	1 (3.9%)
Lymph edema upper limb	1 (3.9%)	1 (3.9%)	0
Neoadjuvant chemotherapy [*n*]	24 (92.3%)	13 (50%)	11 (42.3%)
Preoperative imaging [*n*]			
Magnetic resonance imaging	21 (80.8%)	13 (50%)	8 (30.8%)
Computed tomography	12 (46.2%)	3 (11.5%)	9 (34.6%)
Spinal angiography	7 (26.9%)	2 (7.7%)	5 (19.2%)
Ultrasound of peripheral nerve	3 (11.5%)	3 (11.5%)	0
Cerebral angiography	1 (3.9%)	1 (3.9%)	0
Median age at surgery [months] (range)	54 (8–229)	54 (8–176)	53 (18–229)

## Data Availability

All data generated or analyzed during this study are included in this article or available upon request. Further inquiries can be directed to the corresponding author.

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
