# Peer review of "Trap-Door Thoracotomy and Clamshell Thoracotomy as Surgical Approaches for Neuroblastoma and Other Thoracic Tumors in Children"

_cancers, 2024, doi:10.3390/cancers16020373_

Round 1
Reviewer 1 Report
Comments and Suggestions for Authors
The authors present in tables a wealth of data concerning intraoperative information, complications, etc. Perhaps some of this data is worth comparing - are there differences between the two types of operations in some aspect, such as the duration of the operation, the frequency of complications, etc
Author Response
Comments and Suggestions for Authors
Comments 1: The authors present in tables a wealth of data concerning intraoperative information, complications, etc. Perhaps some of this data is worth comparing - are there differences between the two types of operations in some aspect, such as the duration of the operation, the frequency of complications, etc.
Response 1: Thank you for your thoughts on comparing intraoperative data. Since trap-door thoracotomy is performed for tumors located at the cervicothoracic junction and clamshell thoracotomy is performed for tumors located in the posterior mediastinum or bilateral dorsal thoracic tumors, the indication for both procedures is different. In addition, different structures are involved due to the different tumor locations, and sparing, resection, or reconstruction of these structures strongly influences the surgical outcome. Therefore, we did not statistically compare intraoperative data between the two approaches.
Reviewer 2 Report
Comments and Suggestions for Authors
Dear Editor and Authors,
Thank you for asking me to review this manuscript titled “Trap-door Thoracotomy and Clamshell Thoracotomy as Surgical Approaches for Neuroblastoma and Other Thoracic Tumors in Children” by Dr. Mayer and colleagues from Germany.
This is a retrospective, single institution study of a small sample of pediatric patients (26 in total) undergoing surgery to excise mediastinal tumors via two different incision approaches a clamshell thoracotomy and a trap-door thoracotomy.
I was quite intrigued to read and review this study as I am personally inclined towards trap-door thoracotomy in my own practice but I have also used clamshell thoracotomy in certain cases.
This is a well written study, albeit limited in scope and information provided. It is concise and well integrated / presented. The language of the manuscript is clear with only minor language editing required and well-illustrated with informative images. I have a small number of comments to make:
Comments:
1. The whole argument presented in lines 57 to 60 is incorrect. Manipulation of the heart intraoperative will not result in cardiac failure but in rapid iatrogenic cardiac arrest as the elevation of the heart will decrease preload and lead to reduced cardiac output!! This is why in such cases cardiopulmonary bypass may need to be used. The authors need to modify this section as suggested above.
2. Was PET CT staging used in these patients and if not what was the reason not to?
3. The authors report that data were recorded from patient hospital records!! How accurate where these records and where there any missing variables? The authors need to comment on the accuracy and completeness of the medical records!
4. The description of the surgical technique should be done in the present tense not past!!
5. How do the authors define major vessels been encased of which 80% were according to their results? If a great vessel is encased and or infiltrated then this constitutes an inoperable case/ may need cardiothoracic support and use of CPB!
6. 27% of patients had tracheal involvement? Was tracheal reconstruction needed?
7. How could the authors claim micro and macro complete resection was achieved when the tumors involved numerous structures!
8. In line 244 in the discussion: this study should not said to be the largest to date given the small number of patients reported
In conclusion, this is an interesting study which is well presented but needs some minor editing prior to publication. Kind regards to all.
Comments on the Quality of English LanguageNeeds some minor English editing.
Author Response
Comments and Suggestions for Authors
Dear Editor and Authors,
Thank you for asking me to review this manuscript titled “Trap-door Thoracotomy and Clamshell Thoracotomy as Surgical Approaches for Neuroblastoma and Other Thoracic Tumors in Children” by Dr. Mayer and colleagues from Germany. This is a retrospective, single institution study of a small sample of pediatric patients (26 in total) undergoing surgery to excise mediastinal tumors via two different incision approaches a clamshell thoracotomy and a trap-door thoracotomy. I was quite intrigued to read and review this study as I am personally inclined towards trap-door thoracotomy in my own practice but I have also used clamshell thoracotomy in certain cases.
This is a well written study, albeit limited in scope and information provided. It is concise and well integrated / presented. The language of the manuscript is clear with only minor language editing required and well-illustrated with informative images. I have a small number of comments to make. In conclusion, this is an interesting study which is well presented but needs some minor editing prior to publication. Kind regards to all.
Comments 1: The whole argument presented in lines 57 to 60 is incorrect. Manipulation of the heart intraoperative will not result in cardiac failure but in rapid iatrogenic cardiac arrest as the elevation of the heart will decrease preload and lead to reduced cardiac output!! This is why in such cases cardiopulmonary bypass may need to be used. The authors need to modify this section as suggested above.
Response 1: Thank you for bringing this to our attention. We agree with this comment and have made changes to the sections that address it. The changes are highlighted in the first paragraph of the Introduction (lines 57-61): “For example, in the case of retrocardiac tumors or tumors infiltrating the pericardium or diaphragm, tumor resection through sternotomy or thoracotomy requires extensive manipulation of the heart with the risk of rapid iatrogenic cardiac arrest due to decreased cardiac preload and subsequent reduced cardiac output”, and the fifth paragraph of the Discussion (lines 315 - 318): “For retrocardiac tumors infiltrating the pericardium or diaphragm, successful resection was possible even in hemodynamically unstable patients because the heart could be retracted cranially after complete dissection from the diaphragm without critically reducing cardiac preload and output.”
Comments 2: Was PET CT staging used in these patients and if not what was the reason not to?
Response 2: Thank you for your question, however, PET-CT staging was not used in any patient in this study. This is because the guidelines of the German Society for Paediatric Oncology and Hematology (GPOH) do not recommend PET-CT as a standard preoperative imaging modality for any of the tumor entities represented in this study.
Comments 3: The authors report that data were recorded from patient hospital records!! How accurate where these records and where there any missing variables? The authors need to comment on the accuracy and completeness of the medical records!
Response 3: Thank you for your comments and advice on reporting data quality. Patient hospital records were complete for demographic and clinical characteristics, as well as surgical outcome, with values for all variables examined in this study. Follow-up data were incomplete for the variable of number of cycles of adjuvant chemotherapy administered in 11 patients, and 8 patients were lost to long-term follow-up because the treating physicians or patients did not respond to requests for follow-up data. A statement regarding patients lost to follow-up has been added to the Results, paragraph one (lines 206-207): “A total of 8 patients were lost to follow-up due to non-response to follow-up requests.”
Comments 4: The description of the surgical technique should be done in the present tense not past!!
Response 4: Thank you for pointing this out. The corresponding second and third paragraphs of the Materials and Methods section have been rewritten in present tense (lines 104-121 and 135-147):
“The trap-door thoracotomy is performed in a modified fashion as previously de-scribed (Figure 1 and Supplementary Video S1) [5–7]. General anesthesia is established with a double-lumen or single-lumen endotracheal tube if the lung apex is involved. The patient is placed in the supine position with a shoulder roll, the arm ipsilateral to the lesion outstretched and the head turned to the contralateral side. A supraclavicular trans-verse incision is made toward the midportion of the sternum, leaving the clavicle intact, and then extended inferiorly in the midline of the sternum to the intercostal space where the inferior end of the tumor extends. The incision is then completed laterally to the anterior axillary line. The intercostal muscles and, if necessary, the pectoralis muscles are di-vided and the internal mammary vessels are transected after ligation. The retrosternal space is dissected bluntly before dividing the sternum with a sternal saw. The osteomuscular flap, the so-called "trap door", can then be lifted laterally to expose the surgical field. Dissection begins with mobilization of the thymus and identification of the major vessels and nerves of the upper mediastinum. After completion of tumor resection, a chest tube is placed and the sternum is reapproximated with steel wires in older patients and non-absorbable sutures in infants. Pericostal sutures are placed to reapproximate the ribs. The pectoralis and sternocleidomastoid muscles and the subcutaneous and dermal layers of the incision are reconstructed with absorbable sutures.”
“The clamshell thoracotomy is performed as a modification of the technique first de-scribed by Bains et al. (Figure 2 and Supplementary Video S2) [8]. General anesthesia is established with a double lumen endotracheal tube. The patient is placed in the supine position with both arms abducted and a roll placed under the mid-chest. A bilateral incision is made below the nipples, sparing the mammary glands, following the course of the intercostal spaces with a connecting transverse incision over the lower part of the ste-num. The intercostal muscles are divided and the internal mammary vessels are transected after ligation. The retrosternal space is dissected bluntly before dividing the sternum with a sternal saw. The cranial osteomuscular flap is lifted with retractors to expose the surgical field. After completion of the tumor resection, bilateral chest tubes are placed and connected with a Y-adapter. The sternum is reapproximated with steel wires in older patients and non-absorbable sutures in infants, and the ribs with pericostal sutures. The subcutaneous and dermal layers of the incision are reconstructed with absorbable sutures.”
Comments 5: How do the authors define major vessels been encased of which 80% were according to their results? If a great vessel is encased and or infiltrated then this constitutes an inoperable case/ may need cardiothoracic support and use of CPB!
Response 5: Thank you for bringing this point up. In the guidelines established by the International Neuroblastoma Risk Group Project (Brisse et al. 2011: Guidelines for Imaging and Staging of Neuroblastic Tumors: Consensus Report from the International Neuroblastoma Risk Group Project; Radiology; doi: https://doi.org/10.1148/radiol.11101352), vascular encasement means that more than 50% of the circumference is in contact with the tumor. We have clarified this in the manuscript under Materials and Methods in the first paragraph by reporting how vascular encasement was defined in our study (lines 93-94): “Vascular encasement was defined as tumor contact with more than 50% of the vessel circumference.”
In our experience, thoracic neuroblastic tumors almost exclusively involve the descending aorta and very rarely the ascending aorta or the aortic trunk. Therefore, only a small number of patients require cardiopulmonary bypass for tumor resection. In all patients in this study, the tumors did not involve the aortic stem and therefore cardiopulmonary bypass was not required.
Comments 6: 27% of patients had tracheal involvement? Was tracheal reconstruction needed?
Response 6: Thank you for your comment. Overall, tumors had contact with the trachea in 3 patients and with the bronchi in 4 patients. Tracheal reconstruction was not required in any patient and bronchial reconstruction with intraoperative bronchoscopy was required in 1 patient due to intraoperative injury. We have added this information in the third paragraph of the Results section (lines 230-231): "Intraoperative bronchial injury occurred in one patient requiring reconstruction and intraoperative bronchoscopy.”
Comments 7: How could the authors claim micro and macro complete resection was achieved when the tumors involved numerous structures!
Response 7: Thank you for your question on how tumor resection status was defined. For resection of neuroblastoma, macroscopic complete resection was defined in line with the definitions by the Neuroblastoma study group of European Society for Paediatric Oncology (SIOPEN) and the German Society for Paediatric Oncology and Hematology (GPOH). Macroscopic complete resection was defined as excision of >95% of visible tumor. In addition in our institution the resection status is confirmed by early postoperative magnetic resonance imaging (Schäfer et al. 2023: Local MRI before and after Tumor Resection in Neuroblastoma: Impact of Residual Disease on Event Free Survival; Journal of Clinical Medicine; doi: 10.3390/jcm12237297). For all other tumor entities for which resection was performed by trap-door thoracotomy or clamshell thoracotomy, microscopic complete resection was reported if resection margins were tumor-free based on histopathologic evaluation.
The manuscript has been updated in the fifth paragraph of the Materials and Methods section (lines 171-174): “For neuroblastoma resections, macroscopic complete resection was defined as resection of >95% of the visible tumor. For all other tumor entities, microscopic complete resection was reported if resection margins were tumor-free based on histopathologic evaluation.”
Comments 8: In line 244 in the discussion: this study should not said to be the largest to date given the small number of patients reported
Response 8: Thank you for your suggestion The sentence in the first paragraph of the Discussion (lines 251-254) has been changed: “This study evaluated trap-door thoracotomy and clamshell thoracotomy for surgical resection of neuroblastoma and other solid cervicothoracic, posterior mediastinal, or bilateral dorsal thoracic tumors in children.”
Reviewer 3 Report
Comments and Suggestions for Authors
Congratulation to this nice paper. The managemnt of these cases are very difficult, The discussion comtains very importatnt practical detailes.
My questions are as follows:
1. 92% of patients received neoadjuvant treatment and the R0 resection is 53%, how can you explain that the recurrence rate is 38.8%?
2. Because of the above mentined problem, is there any trend for immunotherapy in these cases.
3. In the Table 2 the number of the All cases is 24 instead on 26.
Author Response
Comments and Suggestions for Authors
Congratulation to this nice paper. The management of these cases are very difficult. The discussion contains very important practical details.
Comments 1: 92% of patients received neoadjuvant treatment and the R0 resection is 53%, how can you explain that the recurrence rate is 38.8%?
Response 1: Thank you for your comment. Our institution is a high-volume surgical reference for a variety of extracranial solid tumor entities in children and therefore a high proportion of high-risk tumors are treated. In this study, 7 (26.9%) children underwent tumor resection for metastatic disease to the thorax. Local recurrence occurred in 5 of these children and only in 3 children in whom tumor resection was performed as primary treatment. Therefore, we conclude that the oncologic outcome after tumor resection is largely influenced by the initial stage of the disease.
Comments 2: Because of the above mentioned problem, is there any trend for immunotherapy in these cases.
Response 2: Thanks for the interesting comment. Indeed, immunotherapy was administered in 1 patient with intraoperatively detected progression of metastatic osteosarcoma by GD2 antibodies. ALK inhibitors were administered in 1 patient after resection of relapsed neuroblastoma and in 1 patient after relapse of inflammatory myofibroblastic tumor. The first paragraph of the Results section has been updated in the manuscript (line 206): “Neoadjuvant immunotherapy was administered to 3 patients.”
Comments 3: In the Table 2 the number of the All cases is 24 instead on 26.
Response 3: Thank you for the thoughtful correction. The number has been corrected in Table 2 in the Results section (line 238).
Reviewer 4 Report
Comments and Suggestions for Authors
The authors retrospective reviewed the surgically treated cases of pediatric thoracic tumors that required Trap-door thoracotomy or Clamshell thoracotomy. Both approaches are rare for pediatric surgery, and this report included many types of tumors. The utility of the Trap-door and Clamshell approaches for pediatric surgery has been reported previously.
1. For the posterior mediastinal tumors in adults, Clamshell thoracotomy is not the first procedure because the Clamshell thoracotomy is one of the anterior approaches, and the field of vision for posterior mediastinum is not as good as the posterior approach, such as posterolateral thoracotomy or Shaw-Paulson approach.
2. From Figure 1a, the authors' approach is a hemi-clamshell approach, and it is not the trap-door thoracotomy. A trans-manubrium or trap-door approach can be selected if the tumor is located only in the cervicothoracic junction. Were all the cases selecting this hemi-clamshell approach in this analysis?
3. The most essential question is whether this invasive approach affects children's growth. Sometimes, the invasive procedure causes deformity or dysfunction of the surgical site, and it often causes severe issues, especially for children. Unfortunately, there was no information related to this critical topic; hence, no novel information was in this manuscript.
Comments on the Quality of English Language
N/A
Author Response
Comments and Suggestions for Authors
The authors retrospective reviewed the surgically treated cases of pediatric thoracic tumors that required Trap-door thoracotomy or Clamshell thoracotomy. Both approaches are rare for pediatric surgery, and this report included many types of tumors. The utility of the Trap-door and Clamshell approaches for pediatric surgery has been reported previously.
Comments 1: For the posterior mediastinal tumors in adults, Clamshell thoracotomy is not the first procedure because the Clamshell thoracotomy is one of the anterior approaches, and the field of vision for posterior mediastinum is not as good as the posterior approach, such as posterolateral thoracotomy or Shaw-Paulson approach.
Response 1: Thank you for your comment regarding possible surgical approaches for posterior mediastinal tumors. For resection of unilateral posterior mediastinal tumors without contact to vital structures, we also prefer a posterior approach via posterolateral thoracotomy. However, this approach does not allow resection of bilateral dorsal thoracic tumors. In the case of tumor involvement of the esophagus, aorta or superior vena cava, it is our opinion that a clamshell thoracotomy allows better visualization and control of these vital structures compared to a unilateral posterior approach.
Comments 2: From Figure 1a, the authors' approach is a hemi-clamshell approach, and it is not the trap-door thoracotomy. A trans-manubrium or trap-door approach can be selected if the tumor is located only in the cervicothoracic junction. Were all the cases selecting this hemi-clamshell approach in this analysis?
Response 2: Thank you for your comment, which we would like to clarify. The hemiclamshell as described by Lebreton et al. (Lebreton et al. 2009: The hemiclamshell approach in thoracic surgery: indications and associated morbidity in 50 patients; Interdisciplinary Cardiovascular and Thoracic Surgery; https://doi.org/10.1510/icvts.2009.211623) does not include a supraclavicular transverse incision. The long sternal dissection and the dissection of a low intercostal space suggest an analogy to the hemiclamshell approach. However, this variation of the trap-door thoracotomy was chosen in girls to spare the mammary gland. In boys, the intercostal space was dissected above the mammary gland.
Comments 3: The most essential question is whether this invasive approach affects children's growth. Sometimes, the invasive procedure causes deformity or dysfunction of the surgical site, and it often causes severe issues, especially for children. Unfortunately, there was no information related to this critical topic; hence, no novel information was in this manuscript.
Response 3: Thank you for addressing the challenging issue of musculoskeletal disorders in children after open thoracic surgery. The main point of our study was to evaluate the surgical outcome of resection of solid thoracic tumors by trap-door thoracotomy and clamshell thoracotomy in terms of perioperative morbidity and oncologic implications. While we were able to show that both approaches allow for early mobilization and recovery allowing for rapid continuation of adjuvant chemotherapy, the effect of both surgical approaches on pediatric growth is beyond the scope of the current study design. In our opinion, children should be less likely to develop scoliosis after clamshell thoracotomy because the pericostal sutures are placed bilaterally. If patients do develop rip fusions, they are likely to be bilateral, making the development of scoliosis less likely. We see a need for long-term functional outcomes in pediatric solid tumor patients. While current trial protocols do not include long-term functional follow-up, this could be part of updated trial protocols to evaluate the long-term effects of oncologic therapies on quality of life.
Round 2
Reviewer 3 Report
Comments and Suggestions for Authors
Thank you for your answer. Right now the paper is more understandable for me.
Reviewer 4 Report
Comments and Suggestions for Authors
The queries were well addressed. This manuscript will be informative for the specific readers of Cancers as a journal of "methods and technologies development".